# Expression Profiles of *CDKN2A*, *MDM2*, *E2F2* and *LTF* Genes in Oral Squamous Cell Carcinoma

**DOI:** 10.3390/biomedicines10123011

**Published:** 2022-11-23

**Authors:** Karolina Gołąbek, Grzegorz Rączka, Jadwiga Gaździcka, Katarzyna Miśkiewicz-Orczyk, Natalia Zięba, Łukasz Krakowczyk, Maciej Misiołek, Joanna Katarzyna Strzelczyk

**Affiliations:** 1Department of Medical and Molecular Biology, Faculty of Medical Sciences in Zabrze, Medical University of Silesia in Katowice, 19 Jordana Str., 41-808 Zabrze, Poland; 2Department of Forest Management Planning, Poznań University of Life Sciences, 71 C Wojska Polskiego Str., 60-625 Poznan, Poland; 3Department of Otorhinolaryngology and Oncological Laryngology, Faculty of Medical Sciences in Zabrze, Medical University of Silesia in Katowice, 10 C Skłodowskiej Str., 41-800 Zabrze, Poland; 4Clinic of Oncological and Reconstructive Surgery, Maria Sklodowska-Curie National Research Institute of Oncology, 15 Wybrzeże Armii Krajowej Str., 44-102 Gliwice, Poland

**Keywords:** *CDKN2A*, *MDM2*, *E2F2*, *LTF*, oral squamous cell carcinoma, gene expression, cancer

## Abstract

Background: Oral squamous cell carcinoma (OSCC) is one of the most commonly detected neoplasms worldwide. Not all mechanisms associated with cell cycle disturbances are known in OSCC. Examples of genes involved in the control of the cell cycle are *CDKN2A*, *MDM2*, *E2F2* and *LTF*. The aim of this study was to examine the possible association between *CDKN2A*, *MDM2*, *E2F2* and *LTF* mRNA expression and influence on clinical variables. Methods: The study group consisted of 88 Polish patients. The gene expression levels were assessed by quantitative reverse transcription PCR. Results: We found no statistically significant differences in the expression level of *CDKN2A*, *MDM2*, *E2F2* and *LTF* genes in tumour samples compared to margin samples. No association was found between the gene expression levels and clinical parameters, except *E2F2*. The patients with G2 tumours had a significantly higher gene expression level of *E2F2* than patients with low-grade G1 tumours. Conclusions: We have not demonstrated that a change in expression profiles of genes has a significant impact on the pathogenesis of OSCC. It may also be useful to conduct further studies on the use of *E2F2* expression profile changes as a factor to describe the invasiveness and dynamics of OSCC development.

## 1. Introduction

Oral squamous cell carcinoma (OSCC) is one of the most commonly detected neoplasms worldwide. This type of cancer accounted for 2% of all discovered cancers in 2020. In the same year, there were 177,757 deaths [1]. OSCC develops as a result of the individual’s genetic predisposition, deregulation of many genes (such as oncogenes, tumour suppressor genes, DNA repair genes), loss of genome integrity and external factors such as viruses, tobacco smoking and alcohol consumption [2,3]. It seems useful to carry out molecular studies that could deepen our understanding of cell cycle regulation disorders, which could lead to carcinogenesis in OSCC. Examples of genes involved in the control of the cell cycle are *CDKN2A* (cyclin dependent kinase inhibitor 2A), *MDM2* (murine double minute 2), *E2F2* (E2F transcription factor 2) and *LTF* (lactotransferrin). Given the function of these genes, they are crucial for the course of this process. *CDKN2A* plays an important role in cell cycle regulation and the codes for the two proteins p14-ARF and p16-INK4A. p14-ARF interacts with MDM2 and as a result leads to the ubiquitination of p53 [4,5], while p16-INK4A blocks and phosphorylates serine and threonine residues of retinoblastoma (Rb) protein and negatively regulates the cell cycle by obstructing cells in the late G1 phase [5]. *MDM2* is oncogene while encoding E3 ubiquitin ligase [6] and plays a role as a negative factor of many protein such as p53, pRb, P21 or p14 [7]. *E2F2* belongs to the E2F family of transcription factors and it can bind to a Rb protein. It is known that the RB/E2F signalling pathway is associated with an important proliferation control at the G1/S checkpoint [8]. *LTF* encode lactotransferrin. This glycoprotein arrested the G1 to S phase transition and it regulates the expression and function of proteins such as Akt, p21, p19, p27, Cdk2, cyclin E, Cdk4, and cyclin D1 [9,10].

The aim of this study was to examine the possible association between *CDKN2A*, *MDM2*, *E2F2* and *LTF* mRNA expression in the OSCC tumour and margin samples and influence on clinical variables in the Caucasian Polish population.

## 2. Materials and Methods

### 2.1. Patient and Samples

The study group consisted of 88 Polish patients diagnosed with OSCC. The tumour and matching margin specimens were collected following surgical resection at the Department of Otorhinolaryngology and Oncological Laryngology, Faculty of Medical Sciences in Zabrze, Medical University of Silesia in Katowice, Poland and the Maria Sklodowska-Curie National Research Institute of Oncology (formerly known as the Maria Sklodowska-Curie Memorial Cancer Centre and Institute of Oncology), Gliwice, Poland. Tumour staging was based on the American Joint Committee on Cancer (AJCC, version 2007) [11,12] and WHO Classification of Head and Neck Tumours [13]. Margins were verified as being free of cancer by pathologists. All samples after resection were immediately submerged in stabilization solution, RNAlater^®^ (Sigma-Aldrich, Saint Louis, MO, USA), then frozen at −80 °C until RNA extraction. The main inclusion criteria for the OSCC group included a diagnosis of primary tumours and no history of preoperative radio- or chemotherapy. The study was approved by the Bioethics Committee of the Medical University of Silesia (approval no. KNW/022/KB1/49/16 and no. KNW/002/KB1/49/II/16/17) and the Institutional Review Board on Medical Ethics of the Maria Sklodowska-Curie Memorial Cancer Centre and Institute of Oncology in Gliwice (approval no. KB/493-15/08 and no. KB/430-47/13). The study protocol diagram is shown in Figure 1.

### 2.2. RNA Extraction

The tumour and margin tissue samples were homogenized with FastPrep^®^-24 homogenizer (MP Biomedicals, Solon, CA, USA) with ceramic beads Lysing Matrix D (MP Biomedicals, Solon, CA, USA). The RNA was isolated using the RNA isolation kit (BioVendor, Brno, Czech Republic) to the standard instruction. The qualitative and quantitative analysis of all isolated RNA was performed by spectrophotometry in Biochrom WPA Biowave DNA UV/Vis Spectrophotometer (Biochrom, Cambridge, UK).

### 2.3. Complementary DNA (cDNA) Synthesis

Total RNA (5 ng) was reverse-transcribed into cDNA using High Capacity cDNA Reverse Transcription Kit with RNase Inhibitor (Applied Biosystems, Foster City, CA, USA) according to manufacturers’ protocol. The reaction was performed in 20 μL volume containing: 2 μL of 10x Buffer RT; 0.8 μL of 25x dNTP mix (100 mM); 2 μL of 10x RT Random Primers; 1 μL of MultiScribe™ Reverse Transcriptase; 1 μL of RNase inhibitor; 3.2 μL of nuclease free H_2_O and 10 μL of previously isolated RNA. The reaction was carried out in Mastercycler personal (Eppendorf, Hamburg, Germany) with the following thermal profile: 25 °C for 10 min, 37 °C for 120 min, 85 °C for 5 min and 4 °C–∞.

### 2.4. Gene Expression Analysis

The relative genes expression (RQ) analysis was performed by Real Time PCR (qPCR) using TaqMan^TM^ Gene Expression Assays, QuantStudio 5 RealTime PCR System and Analysis Software v1.5.1 (Applied Biosystems, Foster City, CA, USA). The glyceraldehyde-3-phosphate dehydrogenase gene (*GAPDH*) was used as an endogenous control. The comparative threshold cycle (Ct) method 2^−∆∆Ct^ was used to determine the RQ. Seven surgical margin samples were used as a calibrator. The qPCR was performed in a volume of 20 µL using 1 µL of cDNA, 10 µL of TaqMan^TM^ Fast Advanced Master Mix (Applied Biosystems, Foster City, CA, USA), 1 µL of TaqMan^TM^ Gene Expression Assays (Assay ID: Hs00923894_m1 for *CDKN2A*, Assay ID: Hs01066930_m1 for *MDM2*, Assay ID: Hs00914334_m1 for *E2F2*, Assay ID: Hs00918090_m1 for *LTF* and Assay ID: Hs03929097_g1 for *GAPH*), and 8 µL of nuclease free H_2_O (EURx, Gdańsk, Poland). Thermal cycle for all analyzed genes was: 95 °C for 20 s, followed by 40 cycles of 95 °C for 1 s and 60 °C for 20 s.

### 2.5. Statistical Analysis

The distribution of variables was evaluated by the Shapiro-Wilk’s test. The significance between genes expression in the tumour and margin were tested using Spearman’s rank correlation analysis. The Mann–Whitney U test and Kruskal-Wallis test were used for comparison gender, age, smoking, alcohol consumption and genes’ expression. The Kruskal-Wallis test with post hoc tests based on mean-ranks were used to determine the effect of the expression of genes in the tumour and margin on clinical parameters (TNM, G). *p*-Value < 0.05 were considered as statistically significant. The statistical software STATISTICA version 13 (TIBCO Software Inc., Palo Alto, CA, USA) was used to perform all analyses.

## 3. Results

Clinical parameters of the OSCC group are shown in Table 1. In one patient, the histological gradient (G) was not available. The average age was 56.4 years (range: 18–75 years). There were 60 (68.2%) men and 28 (31.8%) women; 69 (78.4%) patients were smokers; 65 (73.9%) reported alcohol consumption; 51 (57.9%) were both smokers and alcohol users.

We found no statistically significant differences in the expression level of *CDKN2A*, *MDM2*, *E2F2* and *LTF* genes in tumour samples compared to margin samples. The relative levels of the expression (RQ values) of *CDKN2A*, *MDM2*, *E2F2* and *LTF* genes are shown in Table 2.

We also assessed the correlation of the expression levels of selected genes with age, gender, smoking, alcohol consumption and clinical characteristics of the patients. We demonstrated several statistically significant relationships. The expression of the *MDM2* gene was negative correlated with age only in men (*p*-Value = 0.03) in tumour samples. A statistically significant negative correlation between patient age and *MDM2* gene expression in tumour was found only in non-smokers (*p*-Value = 0.021). We also showed statistically significant positive correlation between age and *E2F2* expression in the non-drinker in margin (*p*-Value = 0.013). Statistically significant positive correlations occurred in non-drinkers and smokers, referring to the expression of the *CDKN2A* gene (*p*-Value = 0.016) and the *E2F2* gene also in the margin (*p*-Value = 0.001).

No association was found between the gene expression levels, age, gender, smoking, alcohol consumption and clinical parameters (TNM and G) in tumour and margin samples, except *E2F2*. The patients with G2 (moderately differentiated) tumours had a significantly higher gene expression level of *E2F2* than patients with low-grade G1 (well-differentiated) tumours (*p*-Value = 0.031) in tumour samples. These results are shown in Figure 2.

## 4. Discussion

One of the most common malignant tumours of the oral cavity is OSCC [3]. Not all the molecular mechanisms that lead to this type of cancer have yet been elucidated. Therefore, it seems useful to carry out analyses to investigate changes in gene expression, including those related to the regulation of the cell cycle. In this manuscript, we performed an analysis of *CDKN2A*, *MDM2*, *E2F2* and *LTF* expressions in the pathogenesis of OSCC.

We found no statistically significant differences in the expression level *CDKN2A*, *MDM2*, *E2F2* and *LTF* genes in tumour samples compared to margin samples. Interestingly, we demonstrated a statistically significant association with the expression of *CDKN2A* in margin tissue in non-drinkers and smokers. *CDKN2A* expression may be higher in human papillomavirus (HPV) infections [14]. Bearing in mind the above results, it seems useful to investigate the status of HPV infection in our samples. However, the change in the expression of p16 (a protein encoded by CDKN2A) could be not only outcome virus infection but also DNA damage and gene mutation [15]. Tobacco smoke contains mutagenic substances [16], which could be related to our changes in *CDKN2A* expression in smokers. The association between *CDKN2A* expression and HPV infection in oral and cervical cancer demonstrated that HPV acts via the oncoprotein E7, which can bind the Rb protein, resulting in the inhibition of the formation of the Rb complex with *E2F* family of transcription factors (for example E2F2). As a result, free E2F2 in absence of the complex with Rb can increase the expression of *CDKN2A* [17,18,19]. Some studies have demonstrated that changes in the expression, promoter methylation or copy number deletion of *CDKN2A* were relatively common in the development and prognosis of OSCC [14,20,21,22,23,24,25]. There are reports that researchers have observed that decreased *CDKN2A* expression was associated with OSCC [26,27]. There are also studies demonstrating the overexpression of *CDKN2A* in OSCC [28,29]. The Wang and others’ [30] analysis demonstrated that the expression of *CDKN2A* was upregulated in OSCC tissues compared with normal tissues in Oncomine database (*p*-Value < 0.01). Furthermore, mRNA expression of the *CDKN2A* in HNSCC (head and neck squamous cell carcinoma) was upregulated compared with normal tissues in the GEPIA (Gene Expression Profiling Interactive Analysis) database (*p*-Value < 0.01). On the other hand, in the GEPIA database, a high expression of *CDKN2A* was associated with better survival in HNSCC patients (*p*-Value < 0.05) [30]. The different results of the studies can be explained by differences in the pathogenesis of a subset of OSCCs [14].

The next gene we analyzed in our work was the oncogene *MDM2*. The expression of *MDM2* gene was negative correlated with age only in men (*p*-Value = 0.03) in tumour samples. A statistically significant negative correlation between patient age and *MDM2* gene expression in tumour was found only in non-smokers (*p*-Value = 0.021). The lowest expression of MDM2 protein was observed in larynx carcinoma specimens. It is worth adding that the expression of MDM2 by activation of p53 may be caused by DNA damage associated with exposure to tobacco and/or alcohol carcinogens [31]. MDM2 overexpression can be observed in many cancers such as lung cancer, breast cancer, liver cancer, esophagogastric cancer and colorectal cancer. Overexpression of MDM2 is the most common change, especially in patients with lung cancer [6]. Higher expression of this protein has also been observed in HNSCC [32,33]. Change in MDM2 expression is also associated with a poorer prognosis in the squamous cell carcinoma of the tonsillar region [34]. Millon and other [31] noted the overexpressed *MDM2* mRNA and a significant decrease in *MDM* expression in an advanced T stage in tumour samples with HNSCC. The gene expression level was not related with either lymph node involvement or differentiation status of the tumour.

It is known that *E2F2* play a significant role in promoting the cell cycle [8]. Some studies demonstrate increased *E2F2* gene expression in cancers such as ovarian cancer, non-small cell lung cancer, hepatocellular carcinoma and breast cancer [35,36,37,38]. Our study demonstrated a statistically significant positive correlation between age and *E2F2* expression in the non-drinker in margin (*p*-Value = 0.013). Statistically significant positive correlations occurred in non-drinkers and smokers, referring to the expression of *E2F2* gene also in the margin (*p*-Value = 0.001). It is also worth emphasizing that patients with G2 (moderately differentiated) tumours had a significantly higher gene expression level of *E2F2* than patients with low-grade G1 (well-differentiated) tumours. Perhaps these results are related to the fact that *E2F2* can act as both a promoter of cell division and a suppressor. Moreover, the *E2F2* expression may be associated with the *CDKN2A* expression. Another study analyzed *E2F2* mRNA expression levels in the human brain and central nervous system cancers types via Oncomine and GEPIA databases. Researchers have demonstrated the overexpression of *E2F2* in glioblastoma multiforme and low-grade glioma tissues [39]. Li and others [40] documented that *E2F2* variants are predictive biomarkers for recurrence risk in patients with OPSCC (oropharyngeal squamous cell carcinoma) [40]. On the other hand, *E2F2* can also have a function as a suppressor in colon cancer. This mechanism is related to the fact that *E2F2* inhibited the proliferation of cancer cells by inhibiting the expression of survivin and regulating the expression of molecules such as cyclin A2, proto-oncogene C-Myc, minichromosome maintenance complex component 4 and cyclin dependent kinase 2 [41]. In addition, the other study demonstrated a low level of *E2F2* expression in clear cell renal cell carcinoma cancer tissues [42]. It seems that molecules such as small non-coding RNAs—miRNA (microRNA) and long non-coding RNA (lncRNA) could play important roles in the regulation of *E2F* expression. Some authors were suggested that miRNAs (for example microRNA-let-7a, miR-125b, miR-146b-3p, miR-638, miR-31, miR-218, miR-454-3p and miRNA-936) by targeting *E2F2*, could be influenced by cancer progression, overall survival of patients and response to radiochemotherapy. These studies focused on types of cancer such as breast cancer, ovarian cancer, colon cancer, gastric cancer, prostate cancer, hepatocellular carcinoma, non-small cell lung cancer cell, glioma and laryngeal cancer [37,38,41,43,44,45,46,47,48]. It is also known that circ_RPPH1/miR-146b-3p/E2F2 axis can promote the progression of breast cancer [48]. Moreover, circCUL2 suppresses retinoblastoma cells by regulating miR-214-5p/E2F2 axis [49].

It is also known that *LTF* could be used for cancer prognosis [50]. Chiu and others [51] demonstrated that primary tumours of clear cell renal cell carcinoma in women characterized by the downregulation of *LTF* had higher pathologic stages [51]. Another study demonstrated that *LTF* expression was negatively correlated with metastases and T-stage in nasopharyngeal carcinoma (NPC) samples. *LTF*-mRNA expression in NPC tissues was significantly lower than in healthy tissue [52]. In our study, we found no statistically significant differences in the expression of *LTF* genes in tumour samples compared to margin samples and with clinical features of the OSCC. Some studies demonstrate that hypermethylation in promoter regions was the factor that may be involved in expression regulation. Zhang and others [53] demonstrated that the low or absent *LTF* expression in OSCC tissues and TCA8113 cells may be a result of hypermethylation in the promoter regions of LTF, which exerted negative effects on the malignance of OSSC cells [53]. The *LTF* promoter hypermethylation was also reported in prostate cancer [54]. Another factor determined that *LTF* expression could be miRNA (mikroRNA), for example miR-214. Deng and others [55] found that the overexpression of miR-214 reduced *LTF* mRNA levels and protein levels in NPC [55].

In summary, to the best of our knowledge, our study is the first that aimed to determine the association between *MDM2*, *E2F2*, and *LTF* mRNA expression in oral squamous cell carcinoma tumour and margin samples. This is given the idea of Slaughter et al. [56,57] of the initiation of field cancerization being associated with various types of molecular damage, e.g., altered gene expression. A molecular approach to the matching margin can contribute to cancer prevention and control. Our result should be validated on larger and different cohorts to better comprehend the role of *CDKN2A*, *MDM2*, *E2F2*, and *LTF* genes in OSCC. In addition, in vitro/pre-clinical tests with cell lines, and two-dimensional (2D) and three-dimensional (3D) models are required, which could provide valuable information for a better understanding of the oral carcinogenesis process such as invasion and metastasis [58].

## 5. Conclusions

In summary, we have not demonstrated that a change in expression profiles of *CDKN2A*, *MDM2*, *E2F2* and *LTF* genes in tumour and margin tissues has a significant impact on the pathogenesis of OSCC. Therefore, further molecular analysis of other genes involved in the regulation of the cell cycle is required. It may also be useful to conduct further studies on the use of *E2F2* expression profile changes as a factor to describe the invasiveness and dynamics of the OSCC development.

## Figures and Tables

**Figure 1 biomedicines-10-03011-f001:**
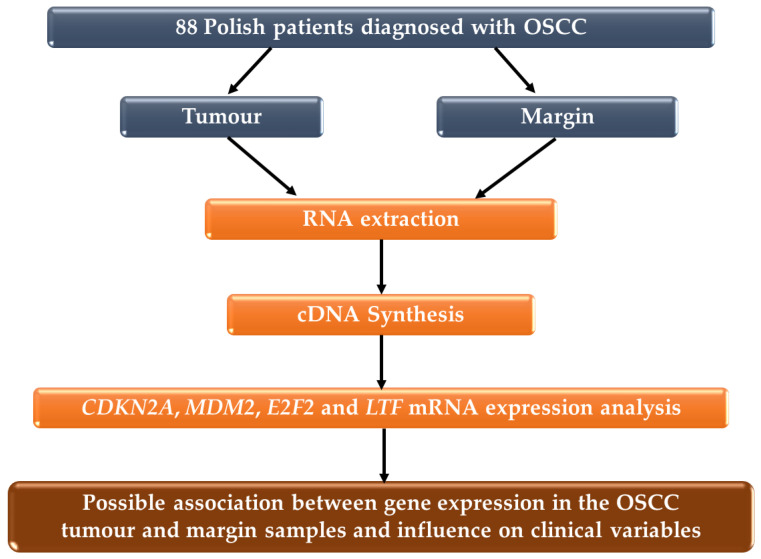
Study protocol diagram.

**Figure 2 biomedicines-10-03011-f002:**
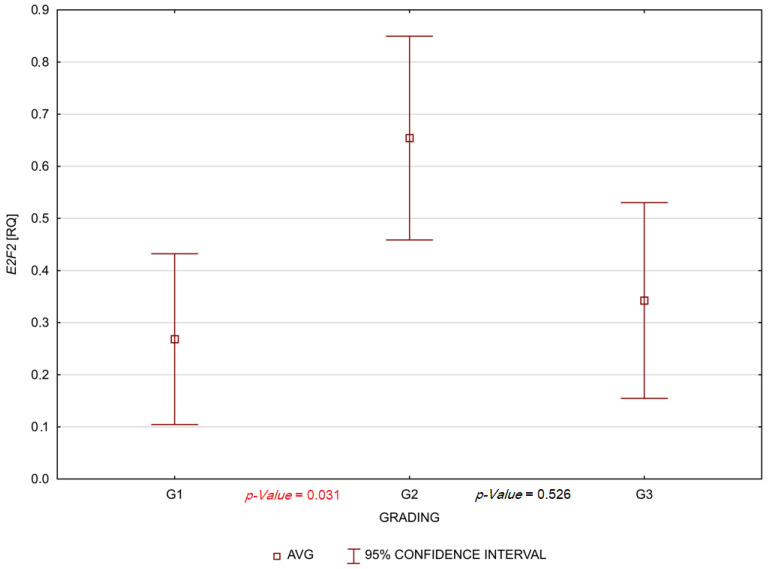
Relative quantification (RQ) values of *E2F2* in the group of patients with OSCC according to G1, G2 and G3 in tumour samples.

**Table 1 biomedicines-10-03011-t001:** Clinical parameters of patients with OSCC.

Clinical Parameters	Patients, n (%)
Histological grading	
G1 (Well differentiated)	14 (16)
G2 (Moderately differentiated)	60 (69)
G3 (Poorly differentiated)	13 (15)
T classification	
T1	8 (9)
T2	24 (27.3)
T3	23 (26.2)
T4	33 (37.5)
Nodal status	
N0	40 (45, 5)
N1	25 (28, 4)
N2	20 (22, 7)
N3	3 (3, 4)

**Table 2 biomedicines-10-03011-t002:** Relative quantification (RQ) values for *CDKN2A*, *MDM2*, *E2F2* and *LTF* genes in tumour vs. margin in patients with OSCC (Spearman’s rank correlation coefficients r, *p*-Value).

Gene	Mean RQ ± SD		r	*p*-Value
	Tumour	Margin		
*CDKN2A*	2.08 ± 4.99	0.45 ± 0.99	−0.031	0.782
*MDM2*	0.7 ± 0.8	0.68 ± 0.39	0.188	0.088
*E2F2*	0.54 ± 0.65	0.5 ± 0.44	0.208	0.061
*LTF*	0.04 ± 0.13	0.13 ± 0.29	0.097	0.378

## Data Availability

The data used to support the findings of this study are available from the corresponding author upon request.

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
