# Peer review of "Expression Profiles of CDKN2A, MDM2, E2F2 and LTF Genes in Oral Squamous Cell Carcinoma"

_biomedicines, 2022, doi:10.3390/biomedicines10123011_

Round 1
Reviewer 1 Report
Title: Expression profiles of CDKN2A, MDM2, E2F2 and LTF genes in oral squamous cell carcinoma
This study aimed to investigate association between CDKN2A, MDM2, E2F2 and LTF mRNA expression in oral squamous cell carcinoma tumour and margin samples. They showed that no statistically significant differences in the expression level CDKN2A, MDM2, E2F2 and LTF genes in tumor samples compared to margin samples. Only E2F2 gene is associated with clinical parameters. The topic and methodology described here is not very innovative and also not interesting. The novelty and scientific value of the findings are rather limited.
There are several concerns that the authors should address in current manuscript:
(1). Four genes (CDKN2A, MDM2, E2F2 and LTF) involved in the regulation of the cell cycle were selected in this study. The authors should describe why they select those genes for their experiments.
(2). Line 45: The sequences of primers and probes should be indicated as a Table in the text.
Reviewer 2 Report
As markers and CDKN2A, MDM2, E2F2, and LTF genes play a vital role in oral squamous cell carcinoma. I found the manuscript well-written and structured. However, there are some areas that require improvements and those need to be addressed before further consideration for this journal. As it is submitted to Biomedicines, not to a clinical Dentistry journal some key biomedical information is required to enhance the readers' interest. Here are my comments needed to be addressed:
- The introduction needs more background information. I recommend adding certain key information on the roles of CDKN2A, MDM2, E2F2, and LTF genes in oral squamous cell carcinoma progression, such as invasion and metastasis. Please include some in vitro/pre-clinical data also to enhance the molecular basis.
- As the study group consisted of 88 Polish patients, please mention the age and gender distribution also. Are there any diseases associated with OSCC? Were they monitored for DM, HTN, or any metabolic disorders? This information is vital for gene expression-related marker studies.
- In figure 1, please show the P values between the points, especially between G1 and G3.
- After discussion, I strongly suggest adding the strengths and limitations of the study. Also, cite some articles to show further work needed to be done in the conclusion using the primary cells if possible with the robust 3D cell culture models for the detailed analysis. I recommend citing these works to show the future pre-clinical study settings.
https://doi.org/10.1016/j.yexcr.2018.06.037
Round 2
Reviewer 1 Report
The authors have addressed all my concerns.